# Endoscopic application of mussel-inspired phenolic chitosan as a hemostatic agent for gastrointestinal bleeding: A preclinical study in a heparinized pig model

In Kyung Yoo[1], Keumyeon Kim[2], Gawon Song[1], Mi-Young Koh[2], Moon Sue Lee[2], Abdullah Özgür Yeniova[3], Haeshin Lee[2,4]*, Joo Young Cho[5]*

1 Department of Gastroenterology, Cha Bundang Medical Center, Cha University College of Medicine, Seongnam-si, Gyeonggi-do, Republic of Korea, 2 R&D Center, InnoTherapy Inc., Seoul, Republic of Korea, 3 Division of Gastroenterology, Department of Internal Medicine, Faculty of Medicine, Tokat Gaziosmanpaşa University, Tokat, Turkey, 4 Department of Chemistry, Korea Advanced Institute of Science and Technology, Daejeon, Republic of Korea, 5 Department of Gastroenterology, CHA Gangnam Medical Center, College of Medicine, Cha University, Seoul, Republic of Korea

☯ These authors contributed equally to this work.
* haeshin@kaist.ac.kr (HL); cjy6695@naver.com (JYC)

## Abstract

Marine mussels secrete adhesive proteins to attach to solid surfaces. These proteins contain phenolic and basic amino acids exhibiting wet adhesion properties. This study used a mussel-inspired hemostatic polymer, chitosan-catechol, to treat gastrointestinal bleeding caused by endoscopic mucosal resection in a heparinized porcine model. We aimed to evaluate the hemostatic efficacy and short-term safety of this wet adhesive chitosan-catechol. We used 15 heparinized pigs. Four iatrogenic bleeding ulcers classified as Forrest Ib were created in each pig using an endoscopic mucosal resection method. One ulcer in each pig was untreated as a negative control (no-treatment group). The other three ulcers were treated with gauze (gauze group), argon plasma coagulation (APC group), and chitosan-catechol hemostatic agent (CHI-C group) each. The pigs were sacrificed on Days 1, 5, and 10, and histological examination was performed (n = 5 per day). Rapid hemostasis observed at 2 min after bleeding was 93.3% (14/15) in the CHI-C group, 6.7% (1/15) in the no-treatment group, 13.3% (2/15) in the gauze group, and 86.7% (13/15) in the APC group. No re-bleeding was observed in the CHI-C group during the entire study period. However, a few re-bleeding cases were observed on Day 1 in the no-treatment, gauze, and APC groups and on Day 5 in the gauze and APC groups. On histological analysis, the CHI-C group showed the best tissue healing among the four test groups. Considering the results, chitosan-catechol is an effective hemostatic material with reduced re-bleeding and improved healing.

## Introduction

Gastrointestinal (GI) bleeding frequently occurs in daily clinical practice and remains to be a potentially life-threatening condition [1, 2]. The most commonly used methods for controlling

**Data Availability Statement:** All relevant data are within the paper and its Supporting Information files.

**Funding:** This study was supported by a grant from the National R&D Program for Cancer Control of the Ministry for Health and Welfare, Republic of Korea (Grant No. HA16C0016 (1631060), H.L.). The funder had no role in study design, data collection and analysis, decision to publish, or preparation of the manuscript. InnoTherapy Inc. only provided financial support in the form of authors salaries (KK, MYK, MSL, and HL) and research materials (CHI-C hemostatic agent), but did not have any additional role in the study design, data collection and analysis, decision to publish, or preparation of the manuscript. The specific roles of these authors are articulated in the 'authors contributions' section.

**Competing interests:** The authors have read the journal's policy and have the following competing interests: Keumyeon Kim and Mi-Young Koh are employees of InnoTherapy Inc. Moon Sue Lee is a founder and owner of InnoTherapy Inc. Haeshin Lee is a co-founder of Innotherapy Inc. and serves as a Chief Technology Officer of InnoTherapy Inc. This does not alter our adherence to PLOS ONE policies on the sharing of data and materials.

GI bleeding are injection (e.g., epinephrine), mechanical (hemoclip), and thermal (argon plasma coagulation or heater probe coagulation) therapies [3]. However, despite recent advances in endoscopic treatment, 10% to 30% of patients encounter treatment failure or re-bleeding [4, 5].

Argon plasma coagulation (APC) is a non-contact thermal method of hemostasis by delivering high-frequency monopolar current through ionized argon gas [6]. The advantages of APC include ease of application, rapid treatment of multiple lesions, and safety due to reduced penetration depth [7]. APC is useful to prevent delayed bleeding in post endoscopic submucosal dissection [8], and it can be used in the endoscopic ablation of early esophageal cancer [9]. Nevertheless, problems such as ulceration, re-bleeding, heat-induced submucosal emphysema, and bowel perforation have been continuously reported [6, 10–13].

To develop an effective hemostatic material for moist living tissues, scientists have studied adhesion mechanisms in marine mussels [14–19]. Mussel secretes an underwater adhesive called mussel adhesive protein rich in phenolic amino acid, 3,4-dihydroxyphenyl-L-alanine (DOPA), and basic amino acid, lysine (~50%) (Fig 1A) [14]. We chemically synthesized an adhesive polymer, chitosan-catechol (CHI-C), to mimic the mussel's underwater adhesion property (Fig 1B) [19]. Chitosan is intrinsically a basic ($-NH_2$) polysaccharide that plays a role similar to that of lysine in the mussel adhesive proteins. Further, the catechol group, a synthetic compound that mimics DOPA, was chemically conjugated to amines of chitosan backbone, and we developed a CHI-C hemostatic agent for GI bleeding (Fig 1B). Chitosan is widely used in biomedical applications due to its excellent biocompatibility and biodegradability. Furthermore, chitosan has been reported to have a strong interaction with the mucin layer of the GI tract by intermolecular hydrogen bonds and electrostatic interactions with negatively charged sialic acid and sulfate residues in mucin [20–22]. The mucoadhesive property of chitosan can be synergistically enhanced by chemical conjugation of catechol. CHI-C shows a mucoadhesive property on the application of an intraperitoneal patch [23] and a possibility for 3D bio-printing materials [24].

So far, hemostasis levels shown by CHI-C treatment have not been compared to existing methods used by clinicians, for example, the APC technique, with an appropriate clinical setting of endoscopic GI bleeding cases. In addition to the hemostatic abilities, the short-term safety concerning re-bleeding and tissue healing ability of CHI-C has not been evaluated. Although CHI-C has shown a significant level of hemostasis, *e.g.*, with no-bleeding syringe needles [25], it was reported in a small animal mouse model. The hemostatic efficacy of CHI-C for a large animal model such as pigs has never been performed. In this study, we developed a heparinized pig model to simulate coagulopathy situations and evaluated the hemostatic efficacy and short-term safety of the wet adhesive CHI-C.

## Materials and methods

### Preparation of CHI-C hemostatic agent

CHI-C was synthesized using standard EDC (1-Ethyl-3-(3-dimethylaminopropyl)-carbodiimide hydrochloride) chemistry as previously reported [26, 27]. Briefly, 3 g of chitosan (100 mPa·s and 80% deacetylated, Heppe Medical Chitosan, Germany) was dissolved in 100 mL of HCl solution (pH 5). 3,4-Dihydroxyhydrocinammic acid (HCA, 2.37 g) and EDC (2.02 g) were dissolved in 25 mL of deionized and distilled water (DDW). The HCA/EDC solution was slowly added to the chitosan solution. During the coupling reaction, the pH value was slowly changed. Thus, it is important to monitor the pH value and to adjust it around 5.0. After 12 hours, the solution was dialysis (molecular weight cut-off: 12,000 to 14,000, SpectraPor, USA) against phosphate-buffered saline (pH 4) for 2 days and then the dialysis solution was changed

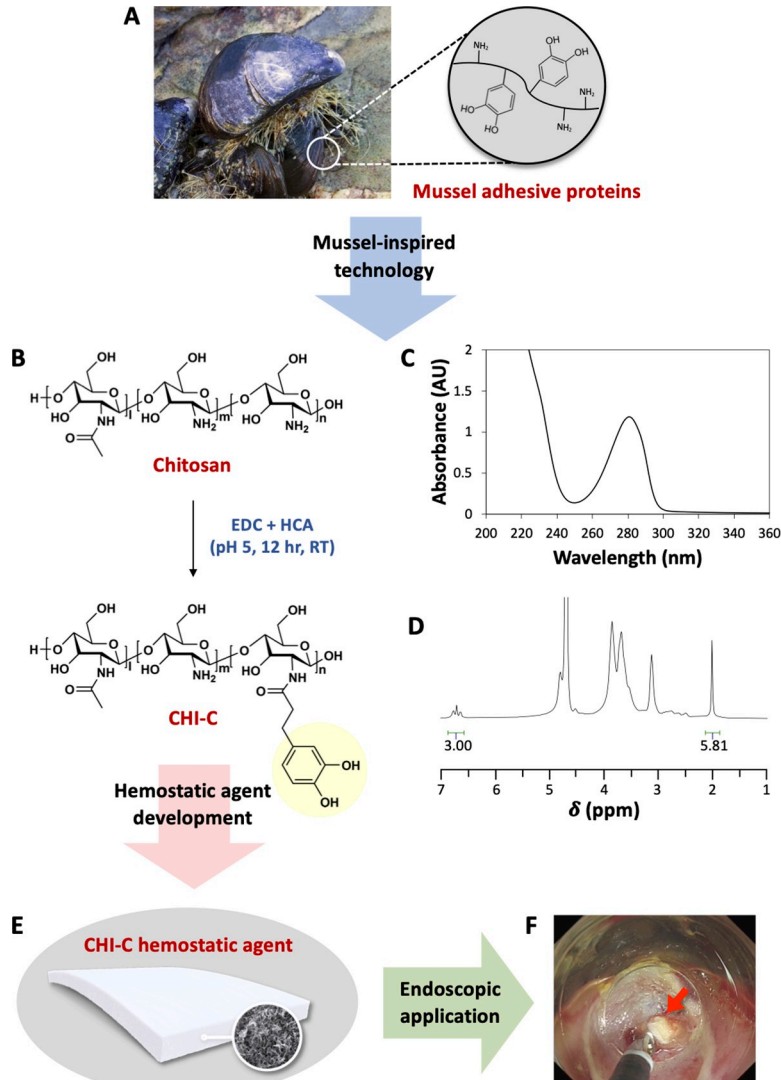

**Fig 1. Mussel adhesive proteins and mussel-inspired chitosan-catechol hemostatic agent.** (A) The coexistence of catechol (DOPA) and amine (lysine) groups in mussel adhesive proteins found in mussel threads is crucial for its underwater adhesion. (B) Synthesis and chemical structure of CHI-C. (C) UV-vis spectrum of the CHI-C solution. (D) $^1$H-NMR spectrum of CHI-C dissolved in $D_2O$. (E) Mussel-inspired hemostatic agent (CHI-C) developed by chemically conjugating catechol groups to amine-rich chitosan, and (F) an endoscopic photograph of the CHI-C hemostatic agent applied to gastric ulcer bleeding.

to DDW for additional 4 hours. Then, the purified solution was freeze-dried. The degree of catechol substitution (DOC) was determined by UV-vis spectroscopy (Hewlett Packard 8453, Agilent) and Nuclear Magnetic Resonance ($^1$H-NMR) spectroscopy (Bruker Avance, 400 MHz, $D_2O$) as previously reported [17, 26]. The CHI-C hemostatic material was prepared by a lyophilization method. CHI-C solution (1 wt%) was dissolved in DDW, and then transferred into polyethylene terephthalate molds. The CHI-C solutions in the molds were lyophilized for 3 days.

## Ethics statement

This study was approved by the Institutional Animal Care and Use Committee of KNOTUS Co., Ltd. (KNOTUS IACUC, approval no.16-KE-200) and conducted in compliance with

Good Laboratory Practice (GLP) regulations (21 CFR Part 58, US FDA). The animals were sedated by intramuscular injection with a mixture of tiletamine/zolazepam (Zoletil®, Virbac, Carros, France) and xylazine (Rompun®, Bayer, Leverkusen, Germany). General anesthesia was maintained by mechanical ventilation with 1% to 2% isoflurane (Forane®, Baxter, Deerfield, IL, USA) in oxygen. If humane endpoints were reached, such as significant weight loss exceeding 20% of initial baseline body weight, lethargy, pyrexia, hypothermia, or behavior that signals severe pain, the animals were sacrificed and excluded from this study. The animals were humanely euthanized by intravenous (IV) injection of suxamethonium chloride (Succicholine Inj® 50 mg/mL, Ilsung Pharmaceuticals, Seoul, Republic of Korea) at a dose of 0.1 mL/kg under deep anesthesia.

## Study design

We performed a non-blinded preclinical study in fifteen minipigs. This study was designed to compare the hemostatic effect and short-term safety of the CHI-C with APC when applied to hemorrhage after endoscopic mucosal resection (EMR) in a heparinized pig. APC was selected as a positive control because the application method is similar to CHI-C. CHI-C and APC require no special skills, allowing an inexperienced endoscopist to perform the hemostatic procedure easily.

The GI bleeding model in heparinized pigs was developed with three consecutive steps (Fig 2A). The first step is to prepare the animal for endoscopic surgery, including fasting, anesthesia, and blood sampling (for baseline). The second step is to perform the endoscopic procedure

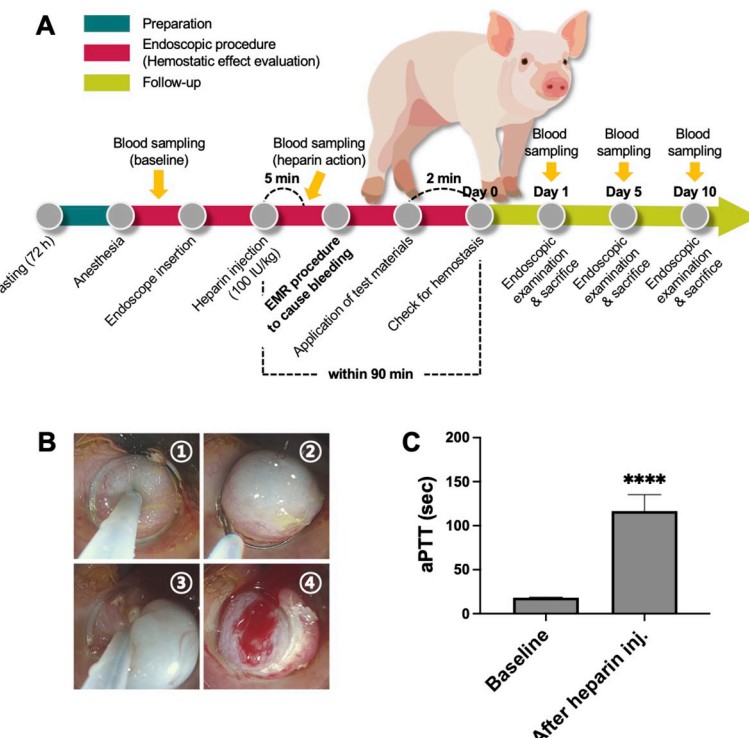

**Fig 2. Study design of EMR-induced GI bleeding in a heparinized pig model.** (A) A schematic diagram of the overall experimental procedure, including animal preparation, endoscopic procedure for hemostatic effect evaluation, and follow-up study for short-term safety evaluation. (B) EMR-C technique in a pig's stomach. (C) aPTT values before (baseline) and after heparin injection. Data were analyzed using a paired t-test. aPTT, activated partial thromboplastin time; inj., injection; ****, $p < 0.0001$.

to assess the hemostatic effects. This step included endoscope insertion, heparin administration (once before EMR procedure), blood sampling (heparin action confirmation), EMR procedure to induce GI bleeding, test material application, and hemostasis evaluation. Each pig received at least four EMR-induced injuries, and four test groups (no-treatment, gauze, APC, and CHI-C) were applied in random order. The location of the ulcers and the information about the applied test materials were recorded. An endoscopy specialist, Dr. I. K. Y., identified each ulcer by comparing it with the records. The last step is a follow-up examination to assess short-term safety for the re-bleeding occurrence and wound healing on Days 1, 5, and 10.

## Experimental group and test materials

Experimental groups were divided into four groups: (1) no-treatment group (as a negative control), (2) gauze group, (3) APC group, and (4) CHI-C group. Fifteen pigs were tested in this study. Each pig had at least four injuries, and all test groups were treated to each ulcer. Five pigs were sacrificed for autopsy and histological analysis on Days 1, 5, and 10, respectively. Gauze (Daehan Medical, Seoul, Republic of Korea) and CHI-C hemostatic agent were cut into 1 × 1 cm-size pieces and folded in half. They were applied to the bleeding site using endoscopic forceps through the endoscope (upper row in Fig 3A and S1 Video). APC (VIO$^{\circledR}$ APC2, Erbe Elektromedizin GmbH, Tübingen, Germany) was applied at a power setting of 40 W with a

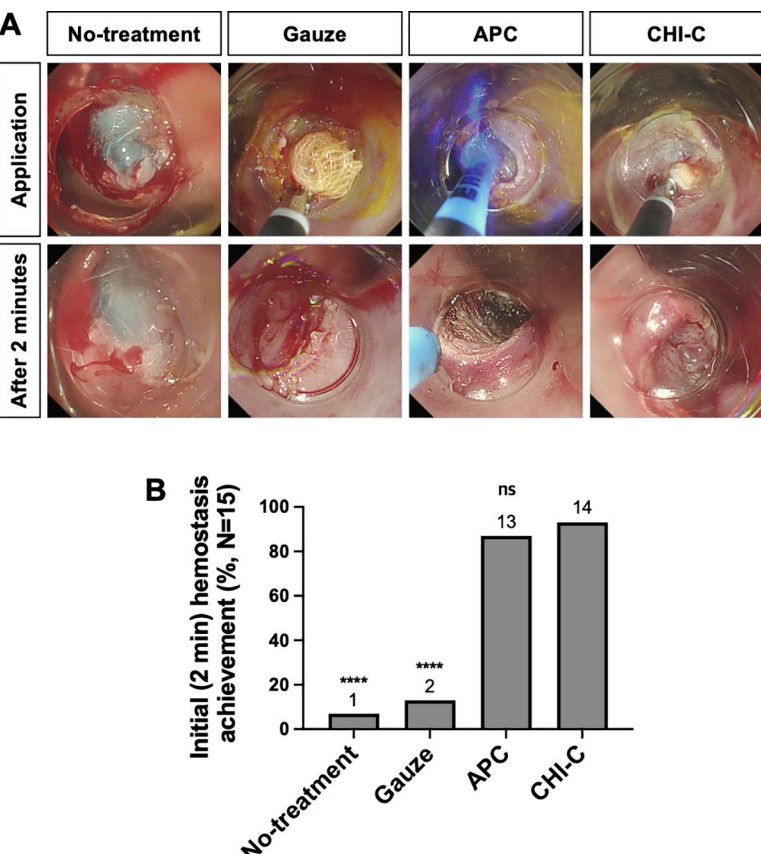

**Fig 3. Initial (2 min) hemostatic effects.** (A) Representative photographic images of the application of each test material (upper panel) and hemostatic status at 2 min after the application (lower panel). (B) Initial hemostasis achievement (%) at 2 min after the application. Data were analyzed using the Mann-Whitney *U* test. ****, $p < 0.0001$; ns, statistically non-significant.

gas flow rate of 2 L/min, and the duration of application was limited to 1 to 3 s at a time (S2 Video).

## Animal preparation

Fifteen male minipigs (Micro-pig®, Medi Kinetics, Pyeongtaek, Republic of Korea) weighing about 50 kg were used. Animals were fasted for 72 h before the endoscopic procedure, with water allowed ad libitum. The animals were sedated by intramuscular injection with a mixture of tiletamine/zolazepam (Zoletil®, Virbac, Carros, France) and xylazine (Rompun®, Bayer, Leverkusen, Germany). General anesthesia was maintained by mechanical ventilation with 1% to 2% isoflurane (Forane®, Baxter, Deerfield, IL, USA) in oxygen. Heart rate, respiratory rate, end-tidal respiration $CO_2$, and oxygen saturation were monitored. After the stabilization of anesthesia, blood sampling for baseline aPTT was performed.

## Cap-assisted EMR induced GI bleeding in a heparinized porcine model

Cap-assisted EMR (EMR-C) was performed using a conventional endoscope (GIF-Q260J; Olympus, Tokyo, Japan) by a gastroenterologist. Heparin sodium (100 IU/kg, Green Cross Corp., Yongin-si, Republic of Korea) was intravenously injected after endoscope insertion. After five minutes, a blood sample for aPTT measurement was collected to confirm heparin action. Since heparin has a half-life of about 90 minutes, all procedures to evaluate the hemostatic effect were completed within 90 minutes after heparin injection [28]. As shown in Fig 2B ①, isotonic saline with methylene blue (3 mL) was injected into the submucosa. A crescent-shaped snare (SnareMaster® snare; loop size 25mm; wire diameter 0.3mm, Olympus medical systems, Tokyo, Japan) was opened and positioned on the internal circumferential ridge of the cap (Fig 2B, ②). Then, the mucosa was suctioned and resected by the electrocautery snare (Fig 2B, ③). The hemostatic evaluation was performed only for bleeding classified as Forrest Ib (Fig 2B, ④). If the bleeding was more severe or milder than Forrest Ib, it was excluded from this study.

## Intraoperative endoscopy to evaluate the hemostatic effect

We evaluated initial hemostasis as an intraoperative outcome. Initial hemostasis was defined as complete cessation of Forrest Ib bleeding 2 min after applying the test material. The APC application duration was limited to 1 to 3 s, and the number of APC applications was counted.

## Follow-up endoscopy

Postoperative endoscopy was performed to evaluate re-bleeding and ulcer healing on Days 1, 5, and 10 after hemostatic effect evaluation. A specialized endoscopist (Dr. I. K. Y) performed the non-blinded assessments. The quality of ulcer healing was quantitatively scored using the Sakita-Miwa classification (Table 1) [29–32]. Sakita-Miwa classification was used to Stage Act-1 was defined as a healing score of 5. Stage Act-2, Heal-1, Heal-2, Scar-1, and Scar-2 were defined as healing scores of 4, 3, 2, 1, and 0, respectively. Further, re-bleeding was evaluated during the ulcer healing evaluation. Re-bleeding was defined as recurrent bleeding when the ulcer was washed with a fine water jet to evaluate ulcer healing endoscopically.

## Histological analysis

After postoperative endoscopy on each predetermined day (Days 1, 5, and 10), five pigs were sacrificed to evaluate histological ulcer healing analysis. If humane endpoints were reached, such as significant weight loss exceeding 20% of initial baseline body weight, lethargy, pyrexia,

**Table 1. Scoring and stage classification of gastric ulcer according to Sakita-Miwa classification.**

| Stage | | Manifestation | Score |
|---|---|---|---|
| Active stage | Act-1 | The surrounding mucosa is edematously swollen and no regenerating epithelium is observed endoscopically. | 5 |
| | Act-2 | The surrounding edema has decreased, the ulcer margin is clear, and a small amount of regenerating epithelium is seen in the ulcer margin. A red halo in the marginal zone and a white slough circle in the ulcer margin are frequently seen. Usually, converging mucosal folds can be followed right up to the ulcer margin. | 4 |
| Healing stage | Heal-1 | The white coating is becoming thin and the regenerating epithelium is extending into the ulcer base. The gradient between the ulcer margin and the ulcer floor is becoming flat. The ulcer crater is still evident and the margin of the ulcer is sharp. The diameter of the mucosal defect is about one-half to two-thirds of that seen in Act-1. | 3 |
| | Heal-2 | The defect is smaller than in Heal-1 and the regenerating epithelium covers most of the ulcer floor. The area of the white coating is about a quarter to one-third of that seen in Act-1. | 2 |
| Scarring stage | Scar-1 | The regenerating epithelium completely covers the floor of ulcer. The white coating has disappeared. Initially, the regenerating region is markedly red. Upon close observation, many capillaries can be seen. This is called "red scar" (Scar-1). | 1 |
| | Scar-2 | In several months to a few years, the redness is reduced to the color of the surrounding mucosa. This is called "white scar" (Scar-2). | 0 |

hypothermia, or behavior that signals severe pain, the animals were sacrificed and excluded from this study. However, none of the tested pigs reached these humane endpoints during our study. The animals were humanely euthanized by intravenous (IV) injection of suxamethonium chloride (Succicholine Inj® 50 mg/mL, Ilsung Pharmaceuticals, Seoul, Republic of Korea) at a dose of 0.1 mL/kg under deep anesthesia. The recovered mucosal tissues were harvested and fixed in 10% neutral buffered formalin, embedded in paraffin blocks, sectioned, and stained with H&E staining following routine procedures. The tissue sections were blindly examined by a histopathologist for the test materials. The quality of ulcer healing was evaluated by observing the regenerated ulcer size and extent/severity of inflammation using a microscope (Olympus BX53, Japan). The extent of inflammation was assessed by dividing the tissue layers into two groups: *i*) lamina propria to submucosa, *ii*) muscularis propria to subserosa. The severity of inflammation was assessed by dividing into two levels according to the updated Sydney system [32]: *i*) none to mild, and *ii*) moderate to severe.

## Statistical analysis

Continuous variables are expressed as mean ± standard deviation (SD). The changes in aPTT values after heparin injection were compared using the paired *t*-test. Repeated measures one-way ANOVA using a mixed-effects model (with the missing data) were performed to compare the healing score of CHI-C over the follow-up endoscopy. Other continuous variables were analyzed using the Mann-Whitney *U* test because of the small sample sizes (non-parametric data). Categorical variables are reported using the proportion (%) and were analyzed using the Fisher's exact test. A *p*-value of less than 0.05 was considered statistically significant. Statistical analyses were performed using Prism version 8.2.0 (GraphPad Software Inc., San Diego, California, USA).

## Results

### Characteristics of CHI-C hemostatic agent

We synthesized CHI-C using standard EDC chemistry with HCA (Fig 1B). The DOC value was determined using the absorbance of catechol at 280 nm by UV-vis spectroscopy (Fig 1C)

in which HCA was used as a standard compound. The DOC in CHI-C was ~8.9% of the total amine group in chitosan. [1]H-NMR was also used to confirm the DOC value by comparing the ratio of the relative integral values for the catechol protons (3H at 6.67 ppm) to those of N-acetyl group protons (3H at 1.95 ppm) in chitosan backbone (80% deacetylated) (Fig 1D). The DOC in CHI-C calculated by [1]H-NMR spectrum was ~10.3%. The error in DOC values might be originated from a little deviation of the deacetylation value of chitosan provided by the manufacturer. The CHI-C was dissolved in DDW at a concentration of 1 wt% and lyophilized in rectangular molds, which generated a porous CHI-C hemostatic agent for endoscopic application (Fig 1E and 1F).

## Initial hemostasis achievement

A total of 15 pigs were administered heparin intravenously before the injury. As shown in Fig 2C, the aPTT value was 116.65 ± 18.55 s at 5 min after heparin injection, which was approximately six times longer than that at baseline (18.26 ± 0.57 s, $p < 0.0001$). Forrest Ib bleeding was successfully induced by the EMR procedure, as shown in the no-treatment group's application photograph in Fig 3A. Representative endoscopic images of each test material application are presented in the upper row of Fig 3A. At 2 min after applications, the bleedings were still observed in the no-treatment and gauze groups, whereas hemostasis was achieved in the APC and CHI-C groups (lower row of Fig 3A). Achievement of initial hemostasis at 2 min after treatment was observed in 93.3% (14 of 15) pigs in the CHI-C group, 6.7% (1 of 15) pigs in the no-treatment group, 13.3% (2 of 15) pigs in the gauze group, and 86.7% (13 of 15) pigs in the APC group (Fig 3B). The success or failure of initial hemostasis in the APC group was decided by the bleeding status at 2 min after the APC application, and the number of APC applications was recorded. The number of APC applications in the initial hemostasis failure group was 16 (range: 12–20 times), which was significantly higher than that in the initial hemostasis success group, which was 3 (range: 2–17 times) (S1A Fig). In the cases of initial hemostasis failure in the APC group, continuous bleeding from the cracks between the blood clots formed by APC was frequently observed even when APC was applied several times as shown in S1B Fig (yellow arrows).

## Re-bleeding observation

After the ulcer was washed with a fine water jet to evaluate ulcer healing, re-bleeding was observed endoscopically. Fig 4A shows the incidence of re-bleeding and red arrows in Fig 4B indicate re-bleeding. No re-bleeding was observed in the CHI-C group. In contrast, on Day 1, there were 3 (20%), 1 (6.7%), and 2 (13.3%) cases of re-bleeding observed in the no-treatment group, gauze group, and APC group, respectively. On Day 5, of the remaining 10 pigs, 1 each in the no-treatment group and the APC group showed re-bleeding. No re-bleeding was observed in any group on Day 10.

## Ulcer healing activities I–endoscopic evaluation

Follow-up endoscopy was performed to evaluate the quality of ulcer healing. The representative endoscopic images of each group are shown in Fig 4B. On Day 1 (n = 15), the ulcers were covered with white coatings in all groups. No regenerating epithelium was observed, and the surrounding mucosa was swollen with edema. Notably, the APC-applied ulcers showed necrotic changes in the mucosal layer because of heating (blue arrow in Fig 4B). On Day 5 (n = 10), the white coating became thin, and the edema around the ulcers decreased. A slight amount of epithelium regeneration and a white slough circle around the ulcer margin were observed. In the CHI-C group, the mucosal folds converging with the ulcer margin were

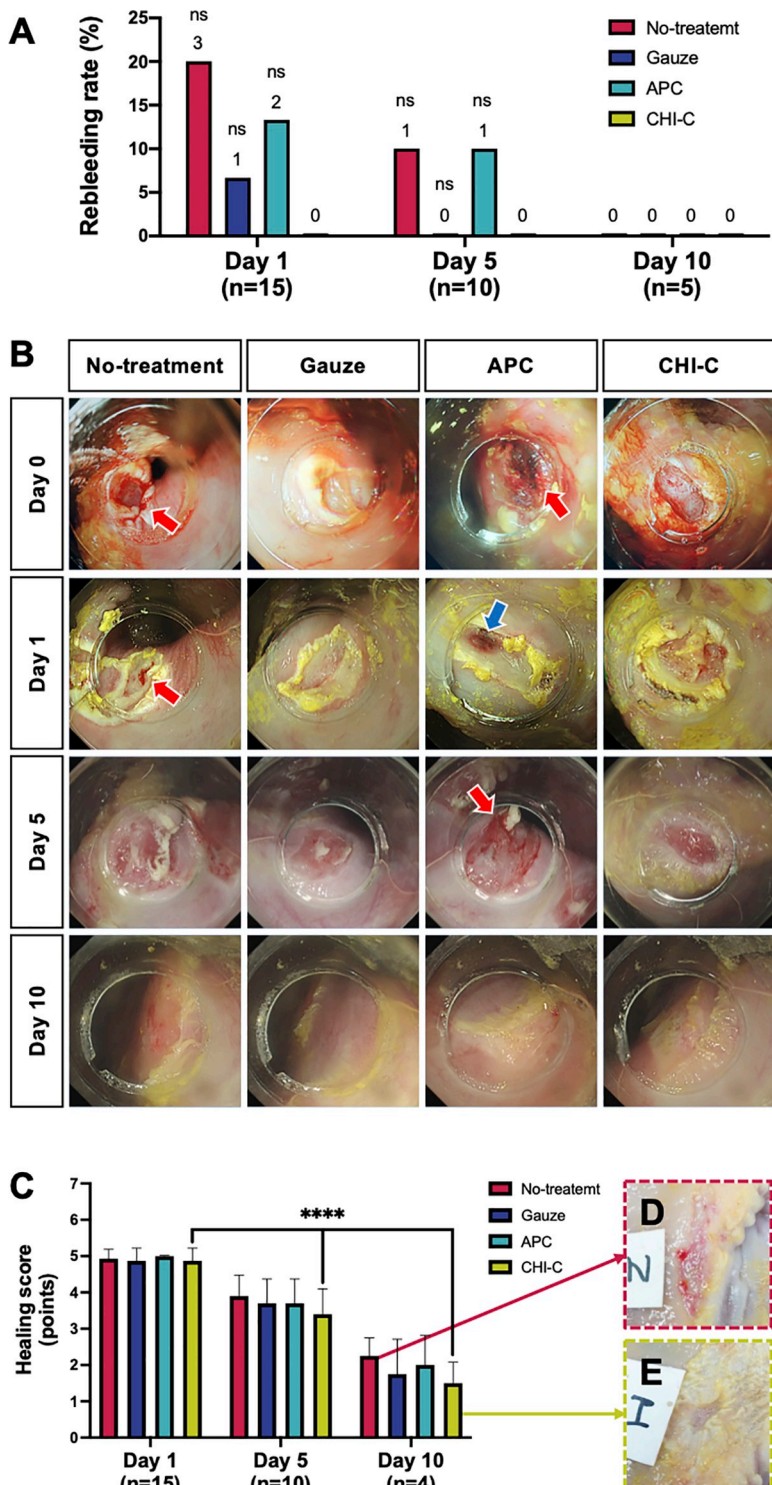

**Fig 4. Rebleeding rate and ulcer healing activity (endoscopic observation).** (A) Rebleeding rate (%) on Days 1, 5, and 10. Data were analyzed using the Fisher's exact test. (B) Representative endoscopic images for the rebleeding and ulcer healing on Days 1, 5, and 10 (red arrow: re-bleeding, blue arrow: necrotic tissue). (C) Healing scores on Days 1, 5, and 10 by endoscopic observation. Data were analyzed by a repeated measures one-way ANOVA using a mixed-effects model with the missing data. (D and E) Representative photographs of ulcers after the autopsy on Day 10. (D) no-treatment group. (E) CHI-C group. ****, $p < 0.0001$; ns, non-significant.

formed. On Day 10 (n = 5), the regenerating epitheliums called "red scar" completely covered the ulcer floor in all groups. To quantitatively assess the quality of ulcer healing, we employed the Sakita-Miwa classification and scored the ulcers (Table 1). Fig 4C shows each group's healing scores, with a high score indicating active ulcers and a low score indicating healed ulcers. The healing scores on Days 1, 5, and 10 in the no-treatment group were 4.93 ± 0.26, 3.90 ± 0.57, and 2.40 ± 0.55, in the gauze group were 4.87 ± 0.35, 3.70 ± 0.67, and 2.00 ± 1.00, in the APC group were 5.00 ± 0.00, 3.70 ± 0.67, and 2.20 ± 0.84, and in the CHI-C group were 4.87 ± 0.35, 3.40 ± 0.70, and 1.80 ± 0.84, respectively. The endoscopic evaluation showed no significant difference in healing scores between the CHI-C group and the other groups. There were no differences statistically, but the ulcer healing manifestation in the CHI-C group (Fig 4E) was better than the no-treatment group (Fig 4D). The healing scores in the CHI-C group decreased significantly over time ($p < 0.0001$ from Day 1 to Day 5, $p < 0.05$ from Day 5 to Day 10).

## Ulcer healing activities II–histological evaluation

We performed histological analysis to confirm the ulcer healing effect of CHI-C. Five ulcers per group were obtained on Days 1 and 5 (n = 5), but four ulcers were obtained on Day 10 (n = 4) because the ulcers in one pig were extensively damaged, so it was difficult to identify each group. Fig 5A shows representative H&E-stained photomicrographs on Days 1, 5, and 10. On Day 1, necrotic cell death and inflammatory cells were observed in a band form at the injury site in all groups. In the APC group, on Day 5, granulation tissues and capillary vessels in the submucosal region were observed (red arrow in Fig 5A). Infiltration of inflammatory cells into the muscularis propria layer was observed in the gauze, APC, and CHI-C groups (asterisks in Fig 5A). In the CHI-C group, on Day 10, most of the damaged tissue was re-epithelialized (blue arrow in Fig 5A).

To quantitatively assess the quality of ulcer healing, we evaluated the relative healing area, depth, and inflammation of ulcers, as shown in Fig 5B. We measured the relative healing area of ulcers in the gauze, APC, and CHI-C groups compared to those of the no-treatment group. The relative healing area in the CHI-C group deteriorated to −92.9% on Day 1 and −88.2% on Day 5. However, on Day 10, the ulcers in the CHI-C group were healed 26.7% more than those in the no-treatment group. The gauze group was also worsened to −108.4% on Day 1 and −58.3% on Day 5, but the ulcers in the gauze group were healed 24.3% on Day 10. In contrast, the APC group was worse during the entire period. The relative healing areas in the APC group deteriorated to −44.4%, −116.9%, and −25.2% on Days 1, 5, and 10, respectively.

We also assessed the quality of ulcer healing by observing the depth (Fig 5C–5E) and inflammation (Fig 5F–5H) of ulcers according to the updated Sydney system [33]. We divided the depth of ulcers into two parts: *i*) lamina propria to submucosa and *ii*) muscularis propria to subserosa. As shown in Fig 5C–5E, in the CHI-C group, three out of five ulcers showed the depth of lamina propria to submucosa layer, and two ulcers showed the depth of muscularis propria to subserosa layer on Day 1. The ulcers in the CHI-C group slightly deteriorated on Day 5. Two out of five ulcers showed the depth of lamina propria to submucosa layer, and three ulcers showed the depth of muscularis to subserosa layer. However, on Day 10, all four ulcers in the CHI-C group showed the depth of lamina propria to submucosa layer, and showed the best ulcer healing effect similar to that observed in the no-treatment group. On Day 10, one out of the four ulcers in the gauze and APC groups showed the depth of muscularis propria to subserosa, and these groups exhibited the worst healing effect among all groups. As shown in Fig 5F–5H, the ulcers in all groups showed the inflammation of moderate-to-severe levels on Days 1 and 5. However, on day 10, two out of four ulcers in the CHI-C

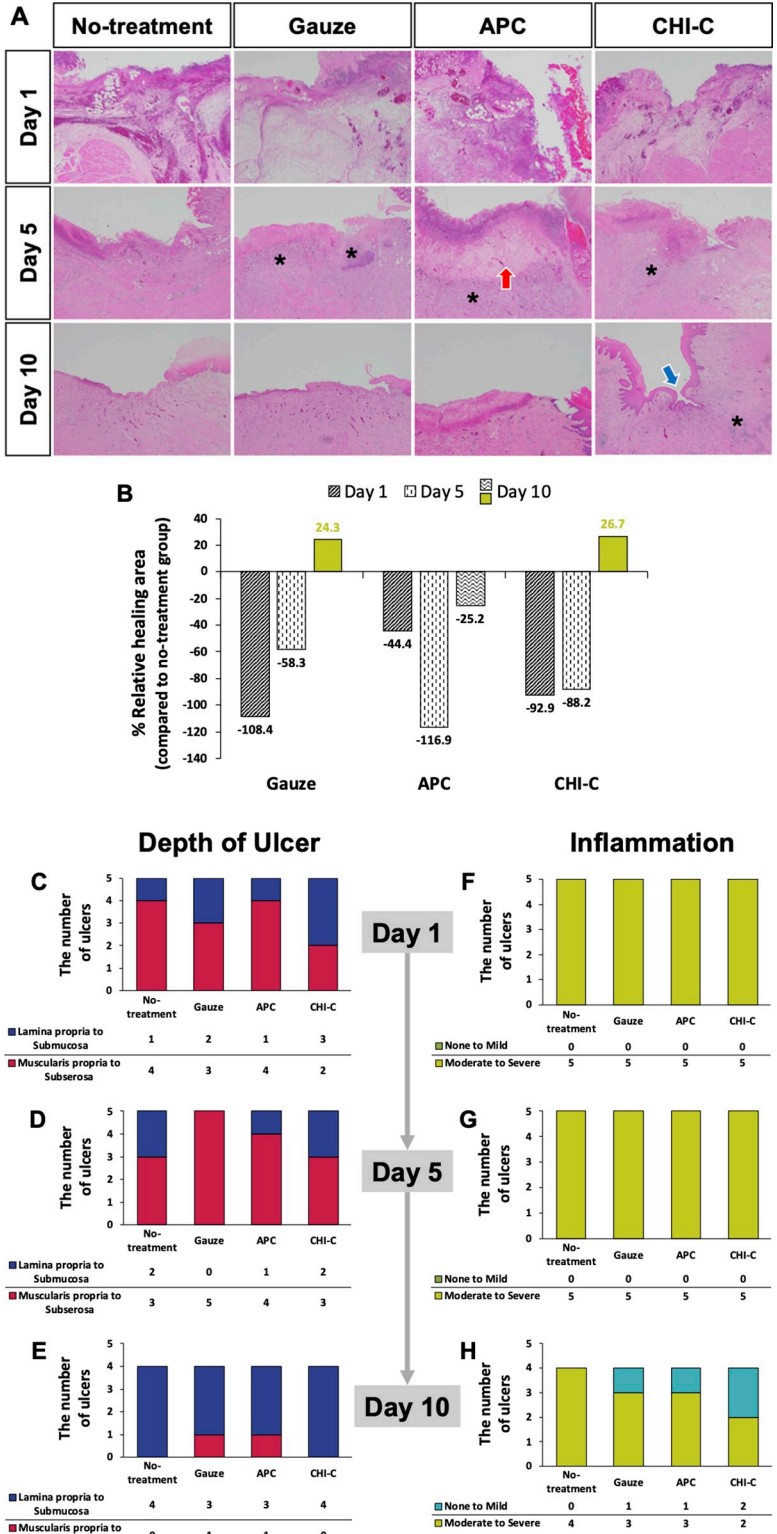

**Fig 5. Histological analysis.** (A) Representative microscopic images on Days 1, 5, and 10 (red arrow: granulation tissue, blue arrow: re-epithelialized layer, asterisk: inflammation in muscular propria). (B) Relative healing area (%) compared to the no-treatment group. (C–E) Ulcer depth on Days 1, 5, and 10. (F–H) Inflammation levels of each group on Days 1, 5, and 10.

group either fully recovered or showed a mild inflammation level, indicating the lowest inflammation level among all groups. In the gauze and APC groups, one out of four ulcers recovered entirely or to a mild inflammation level on Day 10. Whereas, in the no-treatment group, all ulcers showed moderate/mild inflammation levels, indicating the highest level of inflammation on Day 10.

## Discussion

GI bleeding shows the risk of re-bleeding in up to 20% cases [5]. Complete prevention of re-bleeding remains challenging, and re-bleeding is associated with a high mortality rate. Our results showed that CHI-C and APC groups showed similar 2 min hemostasis: 93.3% (14/15) in the CHI-C group and 86.7% (13/15) in the APC group. However, APC-mediated hemostasis causes tissue damage given the principle behind this treatment. Electrical energy is directly transferred to target tissues and produces significant heat in the form of electrical current. Thus, heat-induced denaturation of proteins generates a significant amount of fibrous tissue (red arrow, Fig 5A), which is somewhat related to occurrence of the re-bleeding results shown in Fig 4A and 4B.

In contrast to the significant level of tissue damage caused by APC treatment, the use of CHI-C showed natural healing with no damage of tissues. Recently, we reported tissue healing processes using a CHI-C conjugate for application as intestinal anastomotic sealants [23]. In that study, CHI-C was applied to the partial anastomotic site of the intestine to prevent leakage. CHI-C exhibited strong tissue adhesion as well as rapid tissue healing properties. Tissue adhesion shown by CHI-C is closely associated with its robust binding with proteins [22, 24, 26], and the bound proteins enhance the functions of inflammatory cells such as fibroblasts. Also, hepatectomy clinical trial using CHI-C hemostatic materials showed superior safety in a human level with liver tissue healing [26]. Subsequently, the series of tissue/material reactions promote granulations and re-organization, resulting in accelerated wound healing. Thus, CHI-C is beneficial for rapid healing, which might be responsible for its effective prevention of re-bleeding compared to the prevention efficacy of APC-involved healing.

Another advantage of using CHI-C is that its porous sponge morphology and flexibility enable handling with ease and good accessibility to lesions such as those in the posterior wall of the lesser curvature of the gastric body or posterior wall of the duodenal bulb. Furthermore, CHI-C shows effective bleeding control without thermal insult. APC could induce oozing because coagulation is a spreading pattern rather than a targeting one. Also, APC for GI bleeding can induce unpredictable deep injury and may produce only shallow coagulation insufficient for hemostasis (S1 Fig).

The main limitation of this study is that we only applied the hemostatic methods for an oozing type (Forrest Ib) bleeding. Thus, we cannot generalize the efficacy of CHI-C with other Forrest classifications such as spurting hemorrhage, Forrest Ia. Also, it was difficult to observe differences between groups because the inflammation was still active during the observation period. Studies with other Forrest classifications are necessary in the future for a complete evaluation of the efficacy and safety of hemostasis achieved by CHI-C.

## Conclusions

In conclusion, CHI-C has hemostatic efficacy to stop Forrest Ib GI bleeding in a porcine model receiving anticoagulant therapy. This could be a new endoscopic treatment for GI bleedings. In the previous clinical study, CHI-C hemostatic agent effectively staunched the oozing hemorrhage that cannot be controlled by primary hemostasis during hepatectomy in humans [26]. Future well-designed clinical trials with CHI-C after gastric EMR and

endoscopic submucosal dissection (ESD) should be followed to demonstrate the benefits of its hemostatic effect.

## Supporting information

**S1 Video. A video of applying CHI-C to the bleeding site.**
(MP4)

**S2 Video. A video of applying APC to the bleeding site.**
(MP4)

**S1 Fig. Supporting information of the APC application.** (A) The number of APC applications (times). (B) Photographic images of the failure cases at 2 min after APC application (yellow arrows indicate bleeding).
(TIF)

## Acknowledgments

We would like to thank Ji-Hee Kim for helping with the pig experiment and filming the experiment and the veterinary team of Knotus Co., Ltd. for their help all along with the animal care and experiment supports.

## Author Contributions

**Conceptualization:** Haeshin Lee, Joo Young Cho.

**Data curation:** In Kyung Yoo, Keumyeon Kim, Mi-Young Koh, Haeshin Lee.

**Formal analysis:** In Kyung Yoo, Keumyeon Kim, Abdullah Özgür Yeniova.

**Funding acquisition:** Haeshin Lee, Joo Young Cho.

**Investigation:** In Kyung Yoo, Keumyeon Kim, Gawon Song, Mi-Young Koh, Moon Sue Lee.

**Methodology:** In Kyung Yoo, Keumyeon Kim, Gawon Song, Mi-Young Koh, Moon Sue Lee, Haeshin Lee, Joo Young Cho.

**Project administration:** In Kyung Yoo, Keumyeon Kim, Haeshin Lee, Joo Young Cho.

**Resources:** Moon Sue Lee, Haeshin Lee, Joo Young Cho.

**Supervision:** Haeshin Lee, Joo Young Cho.

**Validation:** In Kyung Yoo, Keumyeon Kim, Gawon Song, Mi-Young Koh, Moon Sue Lee.

**Visualization:** In Kyung Yoo, Keumyeon Kim.

**Writing – original draft:** In Kyung Yoo, Keumyeon Kim.

**Writing – review & editing:** In Kyung Yoo, Keumyeon Kim, Mi-Young Koh, Moon Sue Lee, Abdullah Özgür Yeniova, Haeshin Lee, Joo Young Cho.

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
