## [Decision Letter · Decision Letter 0]

3 Mar 2021

PONE-D-20-38401

Endoscopic application of mussel-inspired phenolic chitosan as a hemostatic agent for gastrointestinal bleeding: a preclinical study in a heparinized pig model

PLOS ONE

Dear Dr. Lee,

Thank you for submitting your manuscript to PLOS ONE. After careful consideration, we feel that it has merit but does not fully meet PLOS ONE’s publication criteria as it currently stands. Therefore, we invite you to submit a revised version of the manuscript that addresses the points raised during the review process.

We look forward to receiving your revised manuscript.

Kind regards,

Panayiotis Maghsoudlou

Academic Editor

PLOS ONE

Journal Requirements:

2. We note that you have provided ethics information within your manuscript. However, we ask that you also provide it in the metadata that you are asked to fill out when you resubmit. This is the section where you are prompted to fill in the title, abstract data availability statement, etc. One of these sections is labelled "Ethics Statement". You have currently written that the information has been included in the manuscript. Please change this to:

'This study was approved by the Institutional Animal Care and Use Committee of KNOTUS Co., Ltd. (KNOTUS IACUC, approval no.16-KE-200) and conducted in compliance with Good Laboratory Practice (GLP) regulations (21 CFR Part 58, US FDA). The animals were sedated by intramuscular injection with a mixture of tiletamine/zolazepam (Zoletil® 95 , Virbac,

5 Carros, France) and xylazine (Rompun® 96 , Bayer, Leverkusen, Germany). General anesthesia was maintained by mechanical ventilation with 1% to 2% isoflurane (Forane® 97 , Baxter, Deerfield, IL, USA) in oxygen. If humane endpoints were reached, such as significant weight loss exceeding 20% of initial baseline body weight, lethargy, pyrexia, hypothermia, or behavior that signals severe pain, the animals were sacrificed and excluded from this study. The animals were humanely euthanized by intravenous (IV) injection of suxamethonium chloride (Succicholine Inj® 50 mg/mL, Ilsung Pharmaceuticals, Seoul, Republic of Korea) at a dose of 0.1 mL/kg under deep anesthesia.'

"This study was supported in part by a grant from the National R&D Program for Cancer

Control, the Ministry for Health and Welfare, Republic of Korea (HA16C0016 (1631060), H.L.)."

"NO authors have competing interests."

We note that one or more of the authors are employed by a commercial company: InnoTherapy Inc..

4.1. Please provide an amended Funding Statement declaring this commercial affiliation, as well as a statement regarding the Role of Funders in your study. If the funding organization did not play a role in the study design, data collection and analysis, decision to publish, or preparation of the manuscript and only provided financial support in the form of authors' salaries and/or research materials, please review your statements relating to the author contributions, and ensure you have specifically and accurately indicated the role(s) that these authors had in your study. You can update author roles in the Author Contributions section of the online submission form.

4.2. Please also provide an updated Competing Interests Statement declaring this commercial affiliation along with any other relevant declarations relating to employment, consultancy, patents, products in development, or marketed products, etc.  

Reviewers' comments:

Reviewer's Responses to Questions

**Comments to the Author**

Reviewer #1: This paper is generally well-written and provides a thorough assessment of the new catechol-chitosan wound patch technology. The paper thoroughly addresses both bleeding as well as wound healing, and the methods for the most part are well done. Thus, I have only a few critiques and requests for clarification:

1. It is unclear why an aPTT was run rather than a PT. While both reflect the effect of heparin on slowing coagulation, PT better reflects the ability of clot to form in tissue, while aPTT reflects the speed of clotting at artificial surfaces. Thus the aPTT is a good measure for what might occur close to the gauze or the CHI-C patch, the PT is a better reflection of the tendency of the injury to bleed.

2. The heparin was given as a bolus, rather than a drip, and thus the aPTT is falling over time, as acknowledged in the paper. Because of this, it is important to denote in what order the interventions were performed (APC, gauze, and CHI-C). Was it randomized or in a specific order? It should be clear if certain interventions faced a more challenging bleed. If so, that could potentially damage the findings for the hemostasis portion of the study.

3. Some of healing data is examined serially in one subject over multiple days. Thus, you should have a statistical method with repeated measures that incorporates a subject variable. From what I can tell, you haven’t done this. For continuous data, such as the % healing measurement, I would used a mixed model because it is better when you have some lost data (which I think you have from one pig). I am not as familiar with non-parametric statistics with repeated measures, so I recommend consulting a statistician to find the ideal repeated measures method for the ranked healing scores.

4. The discussion is a bit unfocused, particularly in its discussion of the healing results. These are scattered over several different paragraphs. Bringing them together and more thoroughly discussing these results in one place would improve the discussion. Thus, I recommend:

P1. In the first paragraph, summarize your findings that CHI-C and APC are the best at stopping bleeding, but that APC causes more tissue injury. Exhaustively discuss the differences in tissue injury here, rather than spread over the other paragraphs.

P2. Then talk about the other advantages you see with CHI-C - accessibility

P3. Then limitations.

Reviewer #2: Interesting study with promising results...

- Further details and clarifications about the CHI-C hemostatic agent preparation, characterization, properties. must be provided (substitution degree, sterilization, etc. ). If all these details are provided in a previously published manuscript, it should be clearly indicated in the text

- I believe that the follow-up endoscopy should have been blinded. Given that APC and CHI-C results are somewhat similar and that the study has other limitations specified by the authors, I would tone down the last sentence in the abstract by replacing outstanding by promising,

- Please clarify if histological analysis was blinded or non-blinded

- Specific comment about the Conclusions: The mention of clinical trials in the last sentence is premature in my opinion, more pre-clinical studies will have to be performed prior to initiate any clinical assessment and this sentence must be revised.

According to PLOS Data policy,

The data should be provided as part of the manuscript or its supporting information, or deposited to a public repository. For example, in addition to summary statistics, the data points behind means, medians and variance measures should be available. If there are restrictions on publicly sharing data—e.g. participant privacy or use of data from a third party—those must be specified.

Some data points are missing as its mostly summary statistics that are provided, it therefore seems that the authors didn't fully comply with this policy

Typos:

Sentence on line 171 is to be revised (was more severe or milder)

While revising your submission, please upload your figure files to the Preflight Analysis and Conversion Engine (PACE) digital diagnostic tool, https://pacev2.apexcovantage.com/. PACE helps ensure that figures meet PLOS requirements. To use PACE, you must first register as a user. Registration is free. Then, login and navigate to the UPLOAD tab, where you will find detailed instructions on how to use the tool. If you encounter any issues or have any questions when using PACE, please email PLOS at figures@plos.org. Please note that Supporting Information files do not need this step

---

## [Author Response · Author response to Decision Letter 0]

17 Apr 2021

The authors are very grateful for the comments from the reviewers and editors. 

The authors are able to answer all raised questions from two reviewers in point-by-point manners successfully. The original reviewer’s comments are in italic, and the authors’ answers are found below for each question. Any changes made in the revised manuscript are marked in red color. 

We believe that these modifications have strengthened the manuscript and hope that the 1st round revised manuscript is suitable for publication in PLOS ONE. Thank you.

---

## [Editor Report · Decision Letter 1]

21 Apr 2021

Endoscopic application of mussel-inspired phenolic chitosan as a hemostatic agent for gastrointestinal bleeding: a preclinical study in a heparinized pig model

PONE-D-20-38401R1

Dear Dr. Lee,

We’re pleased to inform you that your manuscript has been judged scientifically suitable for publication and will be formally accepted for publication once it meets all outstanding technical requirements.

Kind regards,

Panayiotis Maghsoudlou

Academic Editor

PLOS ONE

---

## [Editor Report · Acceptance letter]

6 May 2021

PONE-D-20-38401R1 

Endoscopic application of mussel-inspired phenolic chitosan as a hemostatic agent for gastrointestinal bleeding: a preclinical study in a heparinized pig model 

Dear Dr. Lee:

I'm pleased to inform you that your manuscript has been deemed suitable for publication in PLOS ONE. Congratulations! Your manuscript is now with our production department. 

Kind regards, 

on behalf of

Dr. Panayiotis Maghsoudlou 

Academic Editor

PLOS ONE